# Spatial Profile of Tumor Microenvironment in PD-L1-Negative and PD-L1-Positive Triple-Negative Breast Cancer

**DOI:** 10.3390/ijms24021433

**Published:** 2023-01-11

**Authors:** Liubov A. Tashireva, Anna Yu. Kalinchuk, Tatiana S. Gerashchenko, Maksim Menyailo, Anna Khozyainova, Evgeniy V. Denisov, Vladimir M. Perelmuter

**Affiliations:** 1Laboratory of Molecular Therapy of Cancer, Cancer Research Institute, Tomsk National Research Medical Center, Russian Academy of Sciences, Tomsk 634009, Russia; 2Department of General and Molecular Pathology, Cancer Research Institute, Tomsk National Research Medical Center, Russian Academy of Sciences, Tomsk 634009, Russia; 3Laboratory of Cancer Progression Biology, Cancer Research Institute, Tomsk National Research Medical Center, Russian Academy of Sciences, Tomsk 634009, Russia

**Keywords:** triple-negative breast cancer, tumor microenvironment, immune checkpoint inhibitors, PD-L1, spatial transcriptomic analysis

## Abstract

The problem of finding more precise stratification criteria for identifying the cohort of patients who would obtain the maximum benefit from immunotherapy is acute in modern times. In our study were enrolled 18 triple-negative breast cancer patients. The Ventana SP142 test was used for PD-L1 detection. Spatial transcriptomic analysis by 10x Genomics was used to compare PD-L1-positive and PD-L1-negative tumors. The seven-color multiplex immunofluorescence (by Akoya) was used for the detection of the type of cells that carried the PD1 receptor and the PD-L1 ligand. Using pathway analysis, we showed that PD-L1-positive tumors demonstrate signatures of a cell response to cytokines, among others, and PD-L1-negative tumors demonstrate signatures of antigen presentation. PD-L1-positive and PD-L1-negative tumors have different tumor microenvironment (TME) compositions according to CIBERSORT analysis. Multiplex immunohistochemistry (IHC) confirmed the prevalence of PD1-negative M2 macrophages and PD1-negative T lymphocytes in PD-L1-positive tumors. PD-L1-positive tumors are not characterized by direct contact between cells carrying the PD1 receptor and the PD-L1 ligand. So, the absence of specific immune reactions against the tumor, predominance of pro-tumor microenvironment, and rare contact between PDL1 and PD1-positive cells may be the potential reasons for the lack of an immune checkpoint inhibitor (ICI) effect in triple-negative breast cancer patients.

## 1. Introduction

Because of the high proportion of cases in which therapy is not effective, it is now clear that large financial investments in immunotherapy are not justified. The reasons for this are beyond the PD-L1 status. The use of anti-PD1 pembrolizumab in combination with chemotherapy (nab-paclitaxel, paclitaxel, or gemcitabine–carboplatin) for first-line metastatic triple-negative breast cancer (TNBC) and for patients diagnosed with high-risk early-stage TNBC (anthracyclines plus cyclophosphamide followed by paclitaxel plus carboplatin) as a neoadjuvant treatment and then continued as an adjuvant treatment after surgery was approved in 2020 [1]. Atezolizumab, in combination with chemotherapy in patients with rapidly progressive (unresectable locally advanced or metastatic) triple-negative breast cancer, was approved in March 2019 based on data from the phase 3 IMpassion130 study (NCT02425891) [2]. However, the results published in the *Annals of Oncology* in 2021 show that the study did not improve investigator-assessed progression-free survival (PFS) in patients with PD-L1-positive TNBC. In addition, there was no difference in survival benefit between atezolizumab–paclitaxel and placebo–paclitaxel in the PD-L1-positive population [3]. A more formidable result of therapy is hyperprogressive disease (HPD), which is observed in 10% of advanced TNBC patients treated in IO clinical trials. However, HPD was not found to be associated with worse survival outcomes or known prognostic factors after analysis [4]. Significant differences have not been observed between clinical outcomes according to density in tumor-infiltrating T lymphocytes (TILs) of CD4+/CD8+ lymphocytes, FOXP3+ regulatory T-cells (Tregs), peritumoral and stromal myeloid cells, and PD1-positive and PD-L1-positive immune cells. However, the population of tumor-associated macrophages (TAM), which are characterized by a CD163+CD33+PD-L1+ phenotype and epithelioid morphology, was enriched in patients with hyperprogression [5].

The true mechanisms behind the ineffectiveness of therapy or related hyperprogression are not yet clear. PD-L1 expression is more often associated with the predominance of the Th1-type immune response, but these data are still ambiguous. Since the point of the application of drugs is the immune response in the tumor, it is very important to determine its direction. It is well-known that the immune response in a tumor can both inhibit and support its growth and progression. In this regard, the efficacy of immunotherapy is likely to be related to the type of immunoinflammatory response occurring in the tumor rather than to the presence of the immune response itself. Difficulties in finding efficacy markers may be related to the marked heterogeneity of the tumor microenvironment, even within one tumor tissue [6].

It would be useful to identify the features of the tumor microenvironment of PD-L1-positive tumors and to find something in common or find some molecules that hypothetically could predict the efficacy of therapy. The transcriptional profile of PD-L1-negative tumors will suggest other points for therapeutic intervention or explain why tumor elimination does not occur when immune cells are active. Sequencing of PD-L1-negative tumors has been described to look for correlation with PD-L1 protein levels [7], but there are currently no studies that compare the spatial profiles of gene expression in PD-L1-negative and PD-L1-positive breast tumors and determine their cellular composition. This study is devoted to the investigation of tumor microenvironment heterogeneity and its relationship with PD-L1 expression in breast cancer patients.

## 2. Results

### 2.1. TILs, PD-L1 Assay, and CD274 Expression

We studied tissue samples from six patients (three PD-L1-negative and three PD-L1-positive) by SP142 immunohistochemistry assay and performed spatial transcriptome analysis using Visium 10x technology to evaluate the expression of the gene encoding the PD-L1 protein (Figure 1).

Spots with *CD274* gene expression (encoding the PD-L1 protein) were detected in all the tissue samples, regardless of PD-L1 by the SP142 immunohistochemistry assay. The proportion of *CD274*-expressing spots ranged from 1% to 15% and did not differ in PD-L1-negative and PD-L1-positive patients. The TIL level exceeded 20% in all cases.

### 2.2. Manual Annotation and Initial Data Characterization

All clusters were annotated by a pathologist (V.M.P.) based on the morphology of the associated hematoxylin and eosin image. Clusters were labeled as either one or a combination of the following: normal glands, in situ cancer, invasive cancer, adipose tissue, immune infiltrate, necrosis, tertiary lymphoid structures, or connective tissue (Appendix A). We extracted the genes from the leading edge of the signature (i.e., the genes with the high logFC) and visualized their expression in a UMAP plot (Figure 2).

Both PD-L1-negative and PD-L1-positive patients showed clusters of tumor cells expressing *KRT7*, *KRT8*, and *KRT18*. At the same time, clusters of tumor cells showed the stable expression of stemness genes (*CD44*, *PROM1*) and genes encoding proteins involved in matrix degradation (*MMPs*). Clusters showing the expression of *MHCII* genes were also characteristic of patients in both cohorts. This was more often accompanied by the expression of chemokine genes (*CXCL10* and *CXCL13*).

In addition, we merged data from three PD-L1-negative and three PD-L1-positive samples separately and searched for similar clusters. Only one similar cell cluster was found among PD-L1-negative tumors. Analysis of genes from the leading edge of the signature showed that PD-L1-negative tumors were characterized by *CD163*, *HLA-A*, and *STAB1*. Among PD-L1-positive tumors, only one cluster was the same; the marker genes were *IL1R2*, *CHI3L2*, *CHI3L1*.

### 2.3. Cluster Annotation

In additional analyses, we used gene set enrichment analysis in PD-L1-negative and PD-L1-positive cohorts (Appendix A). Each set of top 100 differentially upregulated genes was subjected to enrichment analysis using the Gene Ontology—Biological Processes (GO:BP) database. Each cluster was then manually annotated using the top enriched pathways and upregulated marker genes as a basis. All cases had clusters that represented features related to tissue structure (GO extracellular structure organization). We identified several GO terms displaying interesting spatial expression patterns related to tissue structure (B-cell activation (GO:0042113), padj = 2.37 × 10^−7^; lymphocyte differentiation (GO:0030098), padj = 2.73 × 10^−6^) in the cluster representing the morphologically tertiary lymphoid structures of one PD-L1-positive patient. Interestingly, in all PD-L1-negative patients, GO terms reflecting antigen presentation, i.e., signs of an initiated immune response, were found in some clusters. In PD-L1-positive cases, GO terms instead characterized the active stimulation of cytokines on cells. In clusters of PD-L1-positive patients, we identified GO terms displaying an inflammation pattern: in two cases, it was GO cellular response to type I interferon (padj = 3.5 × 10^−7^ and padj = 3.7 × 10^−4^) and, in one case, it was cellular response to transforming growth factor beta stimulus (padj = 0.039).

### 2.4. Microenvironment Composition Depending on PD-L1 Tumor Status by Transcriptomic Analyses

The distribution of immune and stromal cell populations of the TME in two cohorts of patients was evaluated with CIBERSORT, which estimates cell type abundance from transcriptomic profiles. The TME composition was studied separately in clusters composed of stromal-only spots and clusters composed of mixed spots containing tumor and immune and stromal cells. In stromal-only spots, significant differences were observed in the relative fraction of T cell gamma delta (higher in PD-L1-negative cases; 0.087 (0.000–0.144) vs. 0.000 (0.000–0.000), *p* = 0.017), resting NK cells (higher in PD-L1-positive cases; 0.056 (0.044–0.090) vs. 0.000 (0.000–0.010), *p* = 0.0004), M0 macrophages (higher in PD-L1-negative cases; 0.037 (0.026–0.055) vs. 0.000 (0.000–0.013), *p* = 0.008), and resting mast cells (higher in PD-L1-positive cases; 0.116 (0.029–0.217) vs. 0.000 (0.000–0.000), *p* = 0.001) (Figure 3A).

In mixed tumor and stromal spots, significant differences were observed in the relative fraction of CD4+ naive T cells (higher in PD-L1-positive cases; 0.019 (0.000–0.049) vs. 0.000 (0.000–0.000), *p* = 0.011), resting NK cells (higher in PD-L1-positive cases; 0.055 (0.015–0.084) vs. 0.000 (0.000–0.070), *p* = 0.004), M0 macrophages (higher in PD-L1-negative cases; 0.071 (0.041–0.118) vs. 0.000 (0.000–0.003), *p* = 0.0001), M2 macrophages (higher in PD-L1-positive cases; 0.000 (0.000–0.043) vs. 0.000 (0.000–0.000), *p* = 0.040), resting mast cells (higher in PD-L1-positive cases; 0.134 (0.022–0.183) vs. 0.000 (0.000–0.000), *p* = 0.00001), and eosinophils (higher in PD-L1-positive cases; 0.007 (0.022–0.183) vs. 0.000 (0.000–0.000), *p* = 0.00001) (Figure 3B).

### 2.5. HLA Gene Set, TFGB1 and 2, CD8A, and CD4 Expression Depending on PD-L1 Status

High MHC-I/II expression and HLA variability both correlate with the response to ICB [8]. Low concentrations of TGFβ in the TME are also associated with the response to ICB [9,10]. Therefore, we evaluated the percentage of spots with the expression of *HLA-DP*, *HLA-DM*, *HLA-DO*, *HLA-DQ*, and *HLA-DR*, as well as *TGFB1* and *TGFB2*, *CD8A*, and *CD4* in tumors.

*HLA-DRA*, the prototypical MHC-II molecule, demonstrated strong expression in all PD-L1-positive samples, as did *HLA-DPA1*. *TGFB1* expression was found in more than 40% of the spots in PD-L1-positive samples (Table 1). In addition, PD-L1-positive tumors were found to have 2–10 times more *CD8A*-expressing spots, 1.5–10 times fewer *TGFB2*-expressing spots, and 1.5–2 times more *CD4*-expressing spots than PD-L1-negative tumors. Overall, the results confirm that PD-L1-negative and PD-L1-positive tumors have similar HLA gene expression profiles but different incidences of T cells.

### 2.6. Microenvironment Composition Depending on PD-L1 Tumor Status by Multiplex IHC Analyses

To determine the differences in TME composition in PD-L1-negative and PD-L1-positive tumors, we performed pan-CK/CD3/CD68/CD163/PD1 labeling by multiplex IHC for FFPE samples. The proportions of PD1-negative and PD1-positive M1 macrophages, M2 macrophages, and lymphocytes in the microenvironment of PD-L1-negative and PD-L1-positive tumors were compared (Figure 4).

Differences were found in the proportions of PD1-negative M2 macrophages and PD1-negative T lymphocyte populations. In PD-L1-positive tumors, the proportion of PD1-negative M2 macrophages was higher than that in PD-L1-negative tumors (34.3 (23.1–46.2) % vs. 12.5 (9.8–17.4) %, *p* = 0.0002) and the proportion of PD1-negative T lymphocytes was higher in PD-L1-positive tumors than in PD-L1-negative tumors (79.9 (74.4–86.7) % vs. 65.9 (52.3–78.9) %, *p* = 0.0197).

### 2.7. CD274 and PDCD1 Gene Co-Expression Pattern

Based on SCRNA-sec, we evaluated the co-expression of the CD274 (encoding the PD-L1 protein) and PDCD1 (encoding the PD1 protein) genes in six breast cancer samples (Table 2).

In all PD-L1-negative cases and in two of three PD-L1-positive cases, the number of spots in which PD-L1 and PD1 genes were co-expressed was extremely minimal. Only in one PD-L1-positive case we found 108 spots of 2700 with the co-expression of PD-L1 and PD1 genes. We also evaluated the expression of the M1 macrophage marker CD68, M2 macrophage marker CD163, cytotoxic lymphocyte marker CD8, B-lymphocyte marker CD20, and T-regulatory lymphocyte marker FoxP3, together with the immune-inhibition signature. We found the following at the same location: M1 macrophages and PD-L1 and PD1 gene co-expression in all cases; M2 macrophages in 4/5 cases; cytotoxic and B lymphocytes in 3/5 cases; T-regulatory lymphocytes in 2/5 cases.

### 2.8. Colocalization of PD-L1-Expressing and PD1-Expressing Cells

To determine which type of cells colocalized in the tumors, we estimated the number of PD-L1-expressing M1 and M2 macrophages in close proximity to other PD1-expressing macrophages or lymphocytes by multiplex IHC for FFPE samples.

The results show that the colocalization of PD-L1-positive M1 macrophages and PD1-positive M1 macrophages was the most frequent of all possible cell colocalizations. Additionally, these cells colocalized with PD1-positive T lymphocytes. When comparing these colocalization variants in PD-L1-negative and PD-L1-positive patients, more PD-L1-positive M1 macrophages and PD1-positive M1 macrophages were shown to colocalize in PD-L1-negative patients than in PD-L1-positive patients (1.00 (0.00–3.25) % vs. 0.00 (0.00–0.00) %, *p* = 0.0354) (Figure 5).

The other colocalization variants occurred with equal frequency and in similar numbers in both patient groups.

## 3. Discussion

Only about 25% of patients with indications for ICI therapy show the expected response. The others, at best, demonstrate stabilization. In this article, we tried to elucidate the characteristics of PD-L1-positive tumors and how these characteristics may affect the efficacy of their response to ICI. Fundamental knowledge suggests that the cause may lie in the polarization of the immune response in the tumor. To this end, we determined the type of immune response in PD-L1-positive tumors and used PD-L1-negative tumors as a comparison. The analysis of biological processes showed that, in all cases we studied, there were immune processes in the tumor. It is known that the TME can be divided into infiltrated–excluded (“cold tumors”) and infiltrated–inflamed (“hot tumors”). The triple-negative breast cancer samples we studied had the same high level of TILs. Existing data on the study of triple-negative breast cancer subtypes allowed to identify a cluster of tumors, which at the epigenome level are characterized by enrichment in immune-related pathways such as response to interferon-beta, the positive regulation of T cell-mediated cytotoxicity, or antigen processing and presentation [11]. At the transcriptional level, our study showed a higher level of enrichment in immune-related pathways in PD-L1-positive tumors, but only in PD-L1-negative patients we observed pathways of antigen presentation but no effector phase of immune response.

Nevertheless, according to the literature, tumor mutation burden levels do not differ in tumors depending on the number of TILs and PD-L1 expression, although trends of higher tumor mutation burden levels in tumors with more TILs and in tumors without PD-L1 expression have been described [12]. The presence of the antigen is not sufficient for the development of an effective immune response. There may be some obstacles for the initiation of a full immune response in PD-L1-negative tumors and, since PD-L1 expression is a physiological accompaniment of the immune response, its absence serves as an additional indicator of immune failure in such tumors.

PD-L1-negative and PD-L1-positive tumors exhibited differences in the types of cells in the microenvironment. Moreover, these differences were spatially contiguous. We examined the transcriptional profiles of spots containing only stroma and spots containing tumor cells and microenvironment cells, that is, the microenvironment adjacent to the tumor cells and located at a distance. We hypothesized that this may differ because some cells need direct contact with the target cell to function while others act through the paracrine influence of cytokines. We found that if we evaluate the differences considering the range of tumor cells, the cell types differ. Moreover, the differences were more pronounced in the microenvironment adjacent to the tumor cells due to CD4+ naïve T cells, M2 macrophages, and eosinophils. All of these cell types were predominant in PD-L1-positive tumors. Interestingly, regarding M2 macrophages, there is evidence that PD-L1 expression not only correlates with the presence of M2 macrophages in the tumor microenvironment [13] but can also be promoted by this cell type [14]. It should be noted that the search for similar clusters for PD-L1-negative and PD-L1-positive tumors allowed us to identify significant signatures of macrophages, which dramatically increases the relevance of these cells as possible predictor markers for immunotherapy.

Proteomic validation showed that M2 macrophages are indeed more abundant in the microenvironment of PD-L1-positive tumors; this difference is at the expense of PD1-negative M2 macrophages. The absence of PD1 expression means that the function of these cells cannot be regulated by PD-L1 in a given tumor and is thus beyond the action of inhibitors. Additionally, the level of PD1-negative T lymphocytes was higher in the microenvironment of PD-L1-positive tumors. PD1 expression is characteristic of mature and depleted lymphocytes [15]. In addition, PD1 suppresses antigen recognition at the lymphocyte-committing stage [16]. Taken together, the increased transcriptomic level of CD4+ naïve T cells in PD-L1-positive tumors, given the absence of PD1 expression, may indicate a higher probability of antigen recognition and the development of an immune response. However, given the elevated levels of M2 macrophages, the development of an immune response may be inhibited due to the polarization of the immune response having a protumor orientation. In this case, the use of inhibitors is likely to have no effect on disease progression. All of these factors may explain the lack of effect of ICI in TNBC.

One of the main focuses of our study is to examine the frequency of direct contact between cells carrying the PD1 receptor and PD-L1 ligand and to determine their type. Transcriptome spatial analysis suggested that the proportion of spots with the colocalization of PD1 and PD-L1 expression was minimal. However, M1 macrophages were detected in each of these spots. This was also confirmed by the IHC study. Indeed, most of the cells in direct contact carrying the PD1 receptor and PD-L1 ligand are M1 macrophages or M1 macrophages and T lymphocytes and their number is greater in PD-L1-negative tumors. PD-L1-positive tumors are not characterized by direct contact between cells carrying the PD-L1 receptor and the PD-L1 ligand. This probably also contributes to the expected efficacy of the therapy.

Of the patients we studied, three are candidates for immunotherapy according to the NCCN criteria. However, by analyzing all tumors, we can assume that PD-L1-negative tumors have the same potency to exhibit immune system effects as PD-L1-positive tumors with successful immunotherapy. Along with this, we found possible reasons for the lack of a therapeutic effect in patients for whom it is indicated. Our results do not completely solve the puzzling lack of effect of PD1/PD-L1 inhibitors in triple-negative breast cancer but do suggest future research directions to find more accurate predictive markers for prescribing immunotherapy and to obtain more clinical benefit.

## 4. Materials and Methods

Ethics statement. The institutional review board (IRB) of the Cancer Research Institute, Tomsk National Research Medical Center approved the study in accordance with good clinical practice guidelines and the Declaration of Helsinki (local IRB approval: 25 August 2020, number 7). All patients signed informed consent to participate in the study.

Patients and tumor samples. The study included 18 breast cancer patients (invasive carcinoma of nonspecific type, triple-negative, stage I-IIB). Six formalin-fixed paraffin-embedded (FFPE) TNBC samples were used for spatial transcriptomic analysis and eighteen samples were used to evaluate the tumor microenvironment composition. None of the patients received neoadjuvant chemotherapy. After surgery, patients received chemotherapy according to the NCCN recommendation.

SP142 PD-L1 assay. The study was performed using the Ventana SP142 test on the Ventana Benchmark Ultra platform according to the protocol recommended by the manufacturer. Evaluation was performed according to the recommendations: negative cases had an absence of any detectable PD-L1 staining or the presence of detectable PD-L1 staining of any intensity in tumor-infiltrating immune cells covering <1% of the tumor area occupied by tumor cells and intratumoral and adjacent peritumoral stroma; positive cases had visible PD-L1 staining of any intensity in immune cells infiltrating the tumor, covering ≥1% of the tumor area occupied by tumor cells and intratumoral and adjacent peritumoral stroma (Ventana Medical Systems Inc. VENTANA PD-L1 (SP142) Assay Interpretation Guide (Oro Valley, AZ, USA)).

RNA Quality Assessment, Sample Preparation, and Library Construction. To assess the quality of FFPE tissue blocks, RNA was extracted from 10 μm-thick FFPE sections using a PureLink FFPE RNA Isolation Kit according to the manufacturer’s recommendation. The quality of RNA was assessed by the mean RNA fragment size and the percentage of total RNA fragments >200 nucleotides using High-Sensitivity RNA ScreenTape on a 4150 TapeStation (Agilent, Santa Clara, CA, USA). FFPE tissue blocks of sufficient quality obtained from 16 breast tumors were sectioned to 5 µm and placed on Visium Spatial Slides. Samples were deparaffinized, H&E stained, imaged, and de-crosslinked according to the demonstrated protocol (CG000409|Rev B, 10x Genomics). Libraries were prepared according to the Visium Spatial Gene Expression Reagent Kits for FFPE (CG000407|Rev C). The concentration of cDNA libraries was measured using the dsDNA High-Sensitivity Kit on a Qubit 4.0 fluorometer (Thermo Fisher Scientific, Waltham, MA, USA) and varied from 0.717 to 7.9 ng/µL. The quality of cDNA libraries was assessed using High-Sensitivity D1000 ScreenTape on a 4150 TapeStation and the peak size varied from 251 to 261 bp (Agilent, Santa Clara, CA, USA).

Sequencing. Libraries were loaded at 1.8 pM and sequenced on a NextSeq 500 System (Illumina, San Diego, CA, USA) using paired-end 150 bp reads according to the following read protocol: read 1, 28 cycles; i7 index read, 10 cycles; i5 index read, 10 cycles; read 2, 50 cycles. The median sequencing depth was 28153 read pairs per spot.

Data processing. The Space Ranger software provided by 10x Genomics (version 1.3.1) was used to perform sample demultiplexing, alignment, tissue detection, fiducial detection, and UMI counting. The pipeline used a probe aligner algorithm for FFPE tissues. The obtained data from Space Ranger were analyzed and visualized within the 10x Genomics Loupe Browser software (version 6.0). Differential gene expression analysis was performed using the ‘FindAllMarkers’ function from Seurat. Cluster-specific spots underwent DEGs analysis against other clusters to each sample. The ‘Aggr’ function for merged tSNE analysis was used to find common cell populations. Log FC and *p*-value adjusted were obtained for all genes. The CIBERSORT algorithm was used for the immune cell analysis based on the gene expression data (log2 values) as described [17].

Multiplex seven-color immunohistochemistry. To evaluate PD1 and PD-L1 expression on T cells and macrophage subsets, as well as the colocalization of cells, immunofluorescence multiplex assays using tyramide signal amplification were used. A multiplex staining protocol was performed using Immunostainer Bond RXm (Leica, Wetzlar, Germany). Sun Z et al. recommended the design and validation of the protocol [18]. The following panel of antibodies was used: anti-human CD3 (Polyclonal, Agilent, Santa Clara, CA, USA), anti-human CD68 (clone PG-M1, Agilent, Santa Clara, CA, USA), anti-human CD163 (clone 10D6, Novocastra, Wetzlar, Germany), anti-human PD1 (clone NAT105, Abcam, Cambridge, UK), anti-human pan-cytokeratin (clone AE1/AE3, Agilent, Santa Clara, CA, USA), and anti-human PD-L1 (clone SP142, Ventana, Oro Valley, AZ, USA). Tissue sections were counterstained with DAPI and mounted on Prolong antifade (Thermo Fisher Scientific, Waltham, MA, USA). The images were acquired on an automated quantitative imaging system, Vectra^®^ 3.0 (Akoya Biosciences, Marlborough, MA, USA). Whole slides were scanned at ×4 magnification and multispectral images of regions of interest (ROIs) were obtained at ×10 magnification. The number of ROIs per slide was 5. Tissue segmentation, cell segmentation, and scoring were performed using InForm^®^ software 2.4.2 (Akoya Biosciences, Marlborough, MA, USA). The following types of cells were identified in the microenvironment: T lymphocytes, CD3+; M1 macrophages, CD68+CD163−; M2 macrophages, CD68−/+CD163+. PD1 and PD-L1 expressions were evaluated in each cell type. In addition, PD1- and PD-L1-expressing cell types in direct contact in the tumor microenvironment were examined. The number of cells was calculated as the percentage of all TILs.

Statistical analyses. Statistical analyses were performed using GraphPad Prism 9. The quantitative data were presented as median and interquartile range (Me (Q1-Q3)). Data were analyzed using the nonparametric Mann–Whitney U-test. *p* values were two-sided and *p* < 0.05 was deemed statistically significant.

## 5. Conclusions

This study showed that PD-L1-negative and PD-L1-positive breast cancer differed in their tumor microenvironment profiles and the cellular composition of the microenvironment depended on the nearness to the tumor cells. PD-L1-positive tumors are not characterized by direct contact between cells carrying PD-1 and PD-L1. These data may explain the low efficacy of immune checkpoint inhibitors in PD-L1-positive patients with triple-negative breast cancer.

## Figures and Tables

**Figure 1 ijms-24-01433-f001:**
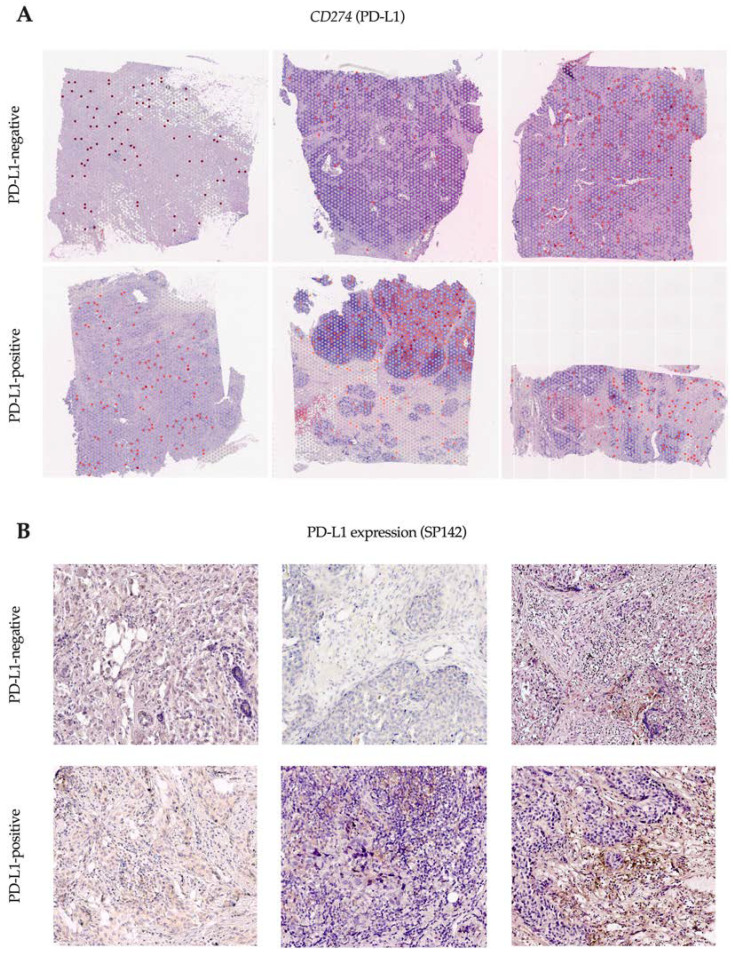
Visium 10x spatial transcriptomic analysis of breast cancer. *CD274* expression in spots indicated as red dots in PD-L1-negative and PD-L1-positive patients (**A**); PD-L1 expression on immune cells (ICs) was assessed by SP142 immunohistochemistry assay in corresponding tissue samples of each patient (**B**). Magnification 200×.

**Figure 2 ijms-24-01433-f002:**
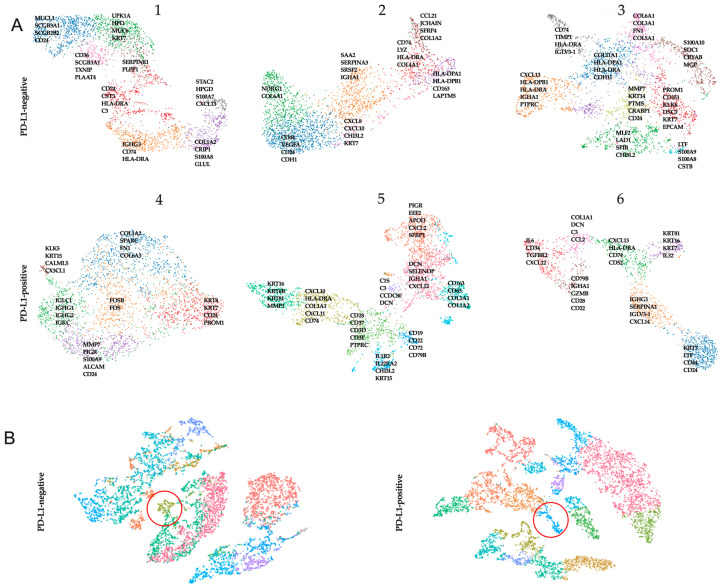
UMAP clustering of tumor samples with the indicated genes of the leading edge of the signature in each cluster in PD-L1-negative and PD-L1-positive patients (**A**). The key genes characteristic of each of the clusters (represented as clouds of dots of different colors) within a single patient (1–6) are presented. Similar cell cluster among PD-L1-negative tumor samples and PD-L1-positive tumor samples (**B**) from merged data, the red circle indicates the similar cell cluster in all patients.

**Figure 3 ijms-24-01433-f003:**
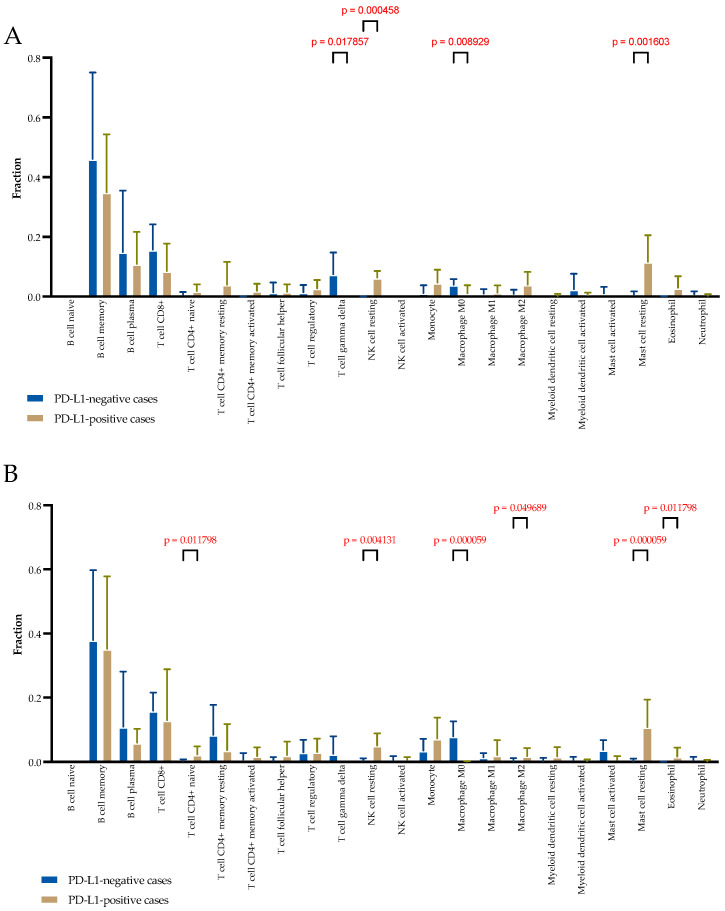
Comparison of the TME composition in stromal-only spots (**A**) and mixed spots (**B**) in the PD-L1-positive cases and PD-L1-negative cases using CIBERSORT analyses.

**Figure 4 ijms-24-01433-f004:**
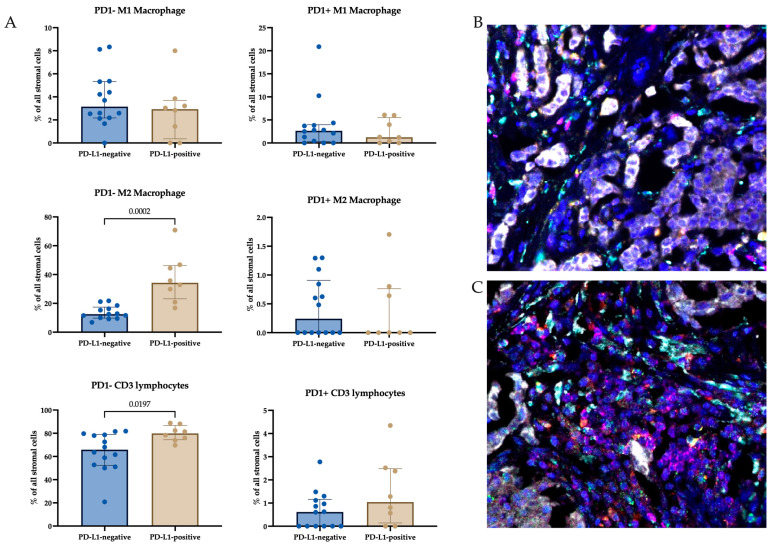
Percentage (**A**) of PD1-negative and PD1-positive M1 and M2 macrophages and T lymphocytes in PD-L1-negative (**B**) and PD-L1-positive (**C**) tumors. M1 (yellow) and M2 (cyan) macrophages and T lymphocytes (purple) were examined using multiplex immunohistochemistry. Magnification 630×.

**Figure 5 ijms-24-01433-f005:**
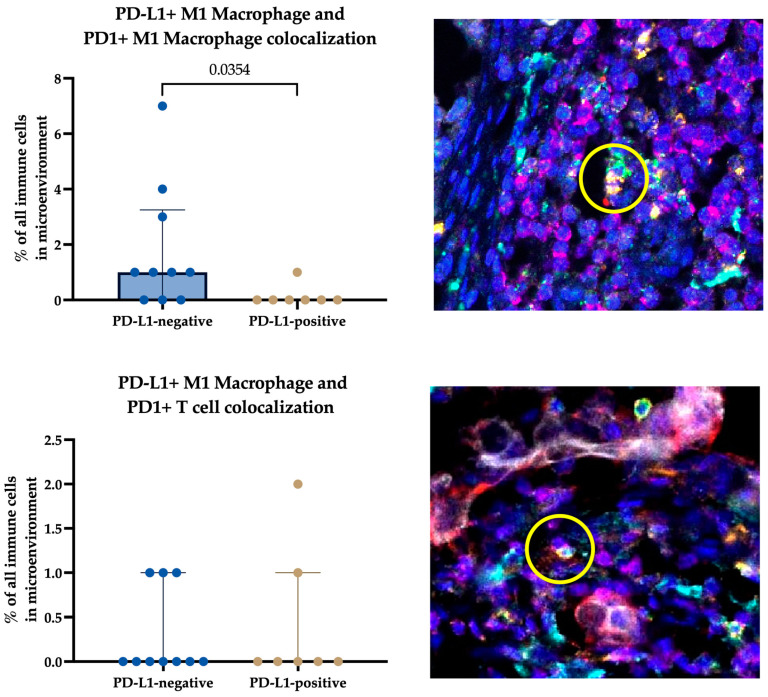
Variants of PD-L1-expressing cells and PD1-expressing cells colocalize in PD-L1-negative and PD-L1-positive tumors. Data are presented as percentages of PD-L1- PD1 cell pairs from all immune cells in the microenvironment. Yellow circle indicates the PD1-PD-L1 interaction in the tumor microenvironment. Multiplex TSA-associated IHC. Magnification 200×.

**Table 1 ijms-24-01433-t001:** HLA gene set, TFGB1 and 2, CD8A, and CD4 expression in spots from PD-L1-negative and PD-L1-positive cases.

Patient		Percent of Spots with Gene Expression, %									
		HLA-DPA1	HLA-DMA	HLA-DMB	HLA-DOA	HLA-DQA1	HLA-DRA	TGFB1	TGFB2	CD8A	CD4
1	PD-L1 negative	96.4	74.9	64.1	18.0	68.5	99.0	50.9	7.6	5.6	32.4
2		37.1	32.1	13.3	2.4	15.6	61.6	35.6	21.0	1.4	11.6
3		92.8	65.3	59.3	24.6	5.8	98.2	65.5	20.2	8.1	46.6
4	PD-L1 positive	92.4	80.4	58.0	28.1	48.4	98.4	40.9	7.7	14.4	31.7
5		88.7	69.0	56.4	26.1	52.8	95.5	57.8	6.8	10.3	56.2
6		94.0	78.8	75.4	45.7	27.9	96.0	65.2	8.3	24.5	67.0

**Table 2 ijms-24-01433-t002:** Percentage of spots with immune cells found at the same location with CD274 and PDCD1 co-expression in PD-L1-negative and PD-L1-positive tumors.

Pts	PD-L1 Status	Percent of Spots with Same *CD274* and *PDCD1* Expression	Percent of Spots with Identified Type of Cells Found at the Same Location %				
			M1	M2	CTL	B cells	Treg
1	negative	0.31	100	77.7	11.1	55.6	33.3
2		0	0	0	0	0	0
3		0.06	100	50	0	0	0
4	positive	0.03	100	0	0	0	0
5		0.07	100	100	50	50	0
6		3.91	91.6	85.2	55.6	71.3	53.7

Note: M1—M1 macrophages, M2—M2 macrophages, CTL—cytotoxic lymphocytes, Treg—T regulatory lymphocytes, Pts—patients.

## Data Availability

The data generated during the current study are available from the corresponding authors upon reasonable request.

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
