# Peer review of "Spatial Profile of Tumor Microenvironment in PD-L1-Negative and PD-L1-Positive Triple-Negative Breast Cancer"

_ijms, 2023, doi:10.3390/ijms24021433_

Round 1
Reviewer 1 Report
1) The SP142 immunostaining in Figure 1 was probably performed from the same specimen, but it is very difficult to tell. If it indicates the presence or absence of PDL-1 positive cells, it should be shown in an enlarged figure. Also, from the FFPE visium image, showing the location of the breast cancer cell population with the breast cancer specific gene plot would certainly reveal PDL-1 positive or negative; for instance, CD274 plots and KRT7, KRT8, KRT18 plots if SPP1 was one of the breast cancer marker. Figure 1 should only clearly show SP142 stained images if it only indicates PDL-1 positive and negative samples.
2) The cluster analysis in each FFPE VISIUM in Figure 2 only shows that there is diversity by specimen. It is difficult to understand the common gene expression described in the text, etc. If each of the three specimens were analyzed together, would there be any difference in the presence or absence of PDL-1?
3) It is difficult to tell whether the bar chart in Figure 3 reflects the x-axis. For example, B cell naive can be understood to mean that it does not appear at all, but it is difficult to know which sample is being compared and what the p-value indicates.
4) Please show the representative IHC images by using FFPE samples in Fig.4. I cannot evaluate whether the bar graphs are true or not. How did the positive cells count? Blind counting?
5) In conclusion, the authors indicated "PD-L1-positive tumors are not characterized by direct contact between cells carrying the PD-1 and PD-L1". In Fig.6, it is very hard to understand the white allows and the conclusion about PD1-PD-L1 interaction. The images are too small and just expand PD1 and PD-L1 images.
Author Response
Dear Editor, Dear Reviewers, we thank you for reviewing our manuscript, for its analysis, and for your evaluation of the study.
Indeed, the tumor microenvironment is a very difficult subject to study and even more so to look for associations with clinical parameters of the disease such as chemotherapy efficacy. This is primarily due to the fact that its cellular composition is very heterogeneous and not the same even within one tissue sample. We showed this when analyzing the cellular composition of spots with direct contact between tumor cells and immune cells and spots containing only immune cells. But the main goal of the article was to find differences in tumor microenvironment in patients with different PD-L1 status. This was to understand whether PD-L1 expression is a sign of a favorable immune response in the tumor, since blocking this immune checkpoint does not always result in a positive therapy effect. Previously published works use either cell lines or animal models or correlation analysis and do not take into account the spatial pattern of the cells. The specificity of our work is that we tried to find physical contact of cell which express PD-L1 and PD1 of immunotherapy application. It turned out that both at the level of mRNA and protein, this is a very rare event, which means that the frequency of positive responses to immunotherapy is highly expected. Our result can be used in the future as an approach to predict the effect of immunotherapy.
Point by point response please find below:
Reviewer 1:
1) The SP142 immunostaining in Figure 1 was probably performed from the same specimen, but it is very difficult to tell. If it indicates the presence or absence of PDL-1 positive cells, it should be shown in an enlarged figure. Also, from the FFPE visium image, showing the location of the breast cancer cell population with the breast cancer specific gene plot would certainly reveal PDL-1 positive or negative; for instance, CD274 plots and KRT7, KRT8, KRT18 plots if SPP1 was one of the breast cancer marker. Figure 1 should only clearly show SP142 stained images if it only indicates PDL-1 positive and negative samples.
Response: Figure 1 demonstrates the presence of PDL-1 gene expression in both PDL-1 positive and negative patients by immunostaining. Figures modified.
2) The cluster analysis in each FFPE VISIUM in Figure 2 only shows that there is diversity by specimen. It is difficult to understand the common gene expression described in the text, etc. If each of the three specimens were analyzed together, would there be any difference in the presence or absence of PDL-1?
Response: Indeed, there is pronounced diversity across samples; the DEGs analysis of the clusters demonstrates this very clearly. The described similar expression patterns involved only marker genes for different cell types, such as tumor cells (KRT7, KRT8, KRT18). Together specimens were analyzed then performed for CIBERSORT analysis, there we took only clusters containing stromal and immune cells and compared in the presence or absence of PD-L1.
3) It is difficult to tell whether the bar chart in Figure 3 reflects the x-axis. For example, B cell naive can be understood to mean that it does not appear at all, but it is difficult to know which sample is being compared and what the p-value indicates.
Response: Naive B-cells were not detected, the p-value indicates differences between PDL-1 positive and PDL-1 negative samples. x-axis and y-axis label was modified. Сell populations are located on the x-axis.
4) Please show the representative IHC images by using FFPE samples in Fig.4. I cannot evaluate whether the bar graphs are true or not. How did the positive cells count? Blind counting?
Response: Images were added to the Figure 4. Counting was performed using the Inform, each image was inspected by a pathologist, using the blind method, and the samples were coded.
5) In conclusion, the authors indicated "PD-L1-positive tumors are not characterized by direct contact between cells carrying the PD-1 and PD-L1". In Fig.6, it is very hard to understand the white allows and the conclusion about PD1-PD-L1 interaction. The images are too small and just expand PD1 and PD-L1 images.
Response: The Figure 6 have been deleted, direct contact between PD-1 and PD-L1 cells represent in Figure 5 and accentuated by a circle.
Reviewer 2 Report
Questions:
1. Is there a typo in the lines 19-20 (PD-L1-positive… and PD-L1-positive)?
2. In the introduction (lines 68-70), references should be given to works in which transcriptomic analysis of PD-L1-negative tumors was performed, if such works are available.
3. It may be worth explaining how the immunohistochemistry results presented in Figure 2b confirm the PD-L1 status of the tumors. At first glance, there are no pronounced differences between positive and negative samples.
4. From the description of section 2.2, it is completely unclear which genes were extracted to obtain these lists. The clarification refers to a change in expression, but what kind of change in expression are authors talking about in such an experiment? Section Materials and methods don’t clarify this extraction.
5. It is practically impossible to compare the clusters in Figure 2 and the numbering of the clusters in Table S1 presenting their detailed description.
6. In lines 101-103 it is necessary to list which genes were found.
7. The text of the article does not describe how the Top100 differentially upregulated genes were identified. What sets were compared with each other to obtain such lists?
8. It is necessary to describe what units of measurement are presented on the Figure 3, how they were calculated, what the bars reflect.
9. Section 2.5 lacks conclusion directly related to its name.
10. In table 2, presented cell types are poorly readable.
11. In the legend for Figure 5, it is necessary to indicate how the presented values ​​were calculated and what the bars reflect.
12. There is strange text in lines 217-219.
13. It is necessary to reveal what the abbreviations TIME and TMB level mean in the discussion section.
Author Response
Dear Editor, Dear Reviewers, we thank you for reviewing our manuscript, for its analysis, and for your evaluation of the study.
Indeed, the tumor microenvironment is a very difficult subject to study and even more so to look for associations with clinical parameters of the disease such as chemotherapy efficacy. This is primarily due to the fact that its cellular composition is very heterogeneous and not the same even within one tissue sample. We showed this when analyzing the cellular composition of spots with direct contact between tumor cells and immune cells and spots containing only immune cells. But the main goal of the article was to find differences in tumor microenvironment in patients with different PD-L1 status. This was to understand whether PD-L1 expression is a sign of a favorable immune response in the tumor, since blocking this immune checkpoint does not always result in a positive therapy effect. Previously published works use either cell lines or animal models or correlation analysis and do not take into account the spatial pattern of the cells. The specificity of our work is that we tried to find physical contact of cell which express PD-L1 and PD1 of immunotherapy application. It turned out that both at the level of mRNA and protein, this is a very rare event, which means that the frequency of positive responses to immunotherapy is highly expected. Our result can be used in the future as an approach to predict the effect of immunotherapy.
Point by point response please find below:
Reviewer 2:
- Is there a typo in the lines 19-20 (PD-L1-positive… and PD-L1-positive)?
Response: Indeed, there is a typo in lines 19-20, and it has been corrected.
- In the introduction (lines 68-70), references should be given to works in which transcriptomic analysis of PD-L1-negative tumors was performed, if such works are available.
Response: Information has been added in the manuscript (ref 7).
- It may be worth explaining how the immunohistochemistry results presented in Figure 2b confirm the PD-L1 status of the tumors. At first glance, there are no pronounced differences between positive and negative samples.
Response: The figure has been modified.
- From the description of section 2.2, it is completely unclear which genes were extracted to obtain these lists. The clarification refers to a change in expression, but what kind of change in expression are authors talking about in such an experiment? Section Materials and methods don’t clarify this extraction.
Response: Indeed, it is more correct to speak not of a change, but of a level of expression. The text has been modified accordingly.
- It is practically impossible to compare the clusters in Figure 2 and the numbering of the clusters in Table S1 presenting their detailed description.
Response: Indeed, we felt that numbering the clusters in the figure would make it harder to read, so in table sets of genes in each cluster was added.
- In lines 101-103 it is necessary to list which genes were found.
Response: Genes were found are listed in text
- The text of the article does not describe how the Top100 differentially upregulated genes were identified. What sets were compared with each other to obtain such lists?
Response: Differential gene expression analysis was performed using the ‘FindAllMarkers()‘ function from Seurat. Cluster-specific spots underwent DEGs analysis against other clusters to each sample. Log FC and p-value adjusted were obtained for all genes.
Explanatory text added to M&M section.
- It is necessary to describe what units of measurement are presented on the Figure 3, how they were calculated, what the bars reflect.
Response: for evaluation level of cells in tumor microenvironment CIBERSORT algorithm was used as described Chen, B.et al. Methods Mol Biol. 2018, 1711, 243-259. doi: 10.1007/978-1-4939-7493-1_12.
Fraction of each cell type was analyzed by the Mann–Whitney U test between the two groups. Data represent as median and interquartile range (Me(Q1-Q3). Text added to M&M section.
- Section 2.5 lacks conclusion directly related to its name.
Response: Section 2.5. was rename.
- In table 2, presented cell types are poorly readable.
Response: Table 2 has been modified
- In the legend for Figure 5, it is necessary to indicate how the presented values ​​were calculated and what the bars reflect.
Response: The legend of Figure 5 is expanded. Data represent as median and interquartile range (Me(Q1-Q3) of the percentages of PD-L1-PD1 cell pairs from all immune cells in the microenvironment.
- There is strange text in lines 217-219.
Response: technical error, text deleted
- It is necessary to reveal what the abbreviations TIME and TMB level mean in the discussion section.
Response: abbreviations “TIME” and “TMB level” deciphered.
Round 2
Reviewer 1 Report
Dear authors,
I am very happy to read the revised manuscripts, and you responded all of my points. I would like to accept it.
However, I have only one simple question. Why did you use "aggr" commands of 10X Chromium analysis and merged tSNE analysis in Loupe browser? It is very easy command to merge six Cellranger data. In Figure.2 or supplemental figure, the merged tSNE analysis will tell a lot of common cell populations, and your data must be improved. If you do not know how to do the "aggregate" comment, please ask 10X chromium team.
Thanks.
Author Response
Dear reviewer, thank you for your positive opinion of the manuscript, with your valuable comments the quality of the manuscript has greatly improved!
We used "aggr" commands of 10X Chromium analysis and merged tSNE analysis for three pdl1-negative and three pdl1-positive samples separately, as you advised. We presented these results in the main body of the article and added in Figure 2. This approach allowed us to find a common cluster of cells characteristic of these tumor groups. We added a corresponding discussion. This greatly improved our results! Thank you for your valuable comments!